# Assessment of the Usefulness of Spectral Bands for the Next Generation of Sentinel-2 Satellites by Reconstruction of Missing Bands

**Jordi Inglada *** , **Julien Michel** and **Olivier Hagolle**

CESBIO, Université de Toulouse, CNES/CNRS/IRD/UPS, 31401 Toulouse, France; julien.michel@cnes.fr (J.M.); olivier.hagolle@cnes.fr (O.H.)
* Correspondence: jordi.inglada@cesbio.eu; Tel.: +33-561-558-676; Fax: +33-561-558-500

**Abstract:** The Sentinel-2 constellation has been providing high spatial, spectral and temporal resolution optical imagery of the continental surfaces since 2015. The spatial and temporal resolution improvements that Sentinel-2 brings with respect to previous systems have been demonstrated in both the literature and operational applications. On the other hand, the spectral capabilities of Sentinel-2 appear to have been exploited to a limited extent only. At the moment of definition of the new generation of Sentinel-2 satellites, an assessment of the usefulness of the current available spectral bands seems appropriate. In this work, we investigate the unique information contained by each 20 m resolution Sentinel-2 band. A statistical quantitative approach is adopted in order to yield conclusions that are application agnostic: multivariate regression is used to reconstruct some bands, using the others as predictors. We conclude that, for most observed surfaces, it is possible to reconstruct the reflectances of most red edge or NIR bands from the rest of the observed bands with an accuracy within the radiometric requirements of Sentinel-2. Removing two of those bands could be possible at the cost of slightly higher reconstruction errors. We also identify mission scenarios for which several of the current Sentinel-2 bands could be removed for the next generation of sensors.

**Keywords:** spectral bands; Sentinel-2; regression; spectral band reconstruction; spectral band selection

## 1. Introduction

The Sentinel-2 constellation constitutes a revolution in remote sensing in terms of data quantity, quality and availability. The high spatial and temporal resolutions of Sentinel-2 [1] have been demonstrated to be crucial for many applications that have been reported in the scientific literature and validated by operational applications covering a wide range of use cases, such as land-cover mapping, snow-extent mapping, biophysical parameter estimation, agriculture monitoring, etc.

Sentinel-2 provides 13 spectral bands with spatial resolutions from 10 m to 60 m and a 5-day revisit cycle.

The particularities of Sentinel-2 with respect to pre-existing comparable systems are:

- in the temporal domain, a systematic acquisition plan (unlike tasked satellites, which acquire scenes on demand) with a high revisit frequency (5 days compared to the 16 days of Landsat);
- in the spatial domain, a higher resolution than Landsat (10 m to 20 m compared to 30 m);
- in the spectral domain, an increased number of bands with respect to both the classical blue, green, red, NIR band set and Landsat (4 visible, 1 NIR, 2 SWIR), with the novelty of 3 red edge (RE) bands, although a lack of thermal band with respect to Landsat.

However, as we show in Section 1.4, very few published works have made full use of the spectral richness of Sentinel-2, and often these uses have not been demonstrated to be the only way to extract the target information.

After 5 years of operation, the work on the new generation of Sentinel-2 satellites (S2NG) has started, and one of the tasks is to identify the set of spectral bands. The question of «Which additional spectral bands could be put on board S2NG?» has to be balanced with the one of the «S2 possible useless bands», that is, the current available bands which could be removed for S2NG. Adding spectral bands to a satellite bears a cost, which could impact the trade-off with other mission requirements, such as temporal revisit needing an additional satellite.

Of course, all current Sentinel-2 bands contain *potentially useful* information, since they sample different intervals of the electromagnetic spectrum, and, except for the pair B8-B8A (see Section 1.1), there is no significant overlap between the different spectral ranges. However, since there exists a high level of redundancy in the underlying observed nature, one can expect high degrees of correlation between the different bands, allowing us to *question the true usefulness* of some bands.

With 5 years of data collection and exploitation, it is now possible to quantitatively assess the usefulness of the different bands on board Sentinel-2. This could be done in terms of the quality of the results of downstream processing (biophysical parameter estimation, land-cover mapping, etc.), but this would need to address a huge number of application domains with experiments and validation data without the guarantee of exhaustivity or chances of replicability.

On the other hand, if we address the problem from the *information content* point of view, we only have to deal with data at the sensor level. We therefore choose to pose the problem as a data reconstruction one: if one band can be reconstructed—within a predefined error margin—from the other bands, it can be removed from the satellite without quantitative loss of information.

One could argue that what matters is the estimation of physical parameters and that imperfect reconstruction of a particular band can have no impact for many applications. This would allow going farther in terms of spectral band removal. We agree with this point of view, but all downstream processing entails the use of (imperfect by construction) models, and the closer we get to the sensor, the most application-independent the conclusions of the study will be.

The aim of this paper is to leverage this interband correlation and assess which bands could be removed from future iterations of the Sentinel-2 constellation with a minimal impact on the usefulness of the acquired data. To do so, we *predict* the reflectances of missing bands with nonlinear regression algorithms that use the other spectral bands as *predictors*. In order to produce results that are representative of real settings and are generalizable, we build a dataset by sampling pixels from Sentinel-2 acquisitions with a wide variety of geographic areas and dates. We therefore take an empirical, data-driven approach.

We choose not to leverage the spatial and the temporal dimensions and carry out a mono-date, pixel-based analysis. We understand that temporal and spatial correlations would reduce the errors in the reconstruction of missing bands. The goal of the work is not to propose an optimal regression algorithm, but rather to show that band reconstruction is possible using regression. The results of this work can be seen as a lower bound in terms of reconstruction quality and therefore encourage the pursuit of further studies.

### 1.1. The Sentinel-2 Spectral Bands

Table 1 gives the name and the central wavelength for each band acquired by Sentinel-2. There are four bands at 10 m resolution: the three usual visible bands (B2–B4) and a wide NIR band (B8). The 20 m resolution bands are three narrow bands in the red edge (B5-B7), one narrow NIR (B8A) and two SWIR bands (B11, B12). Finally, the 60 m resolution bands are aimed at radiometric corrections (B1 for aerosol content estimation, B9 for water vapor and B10 for cirrus detection). Figure 1 illustrates the relative spectral responses of the 10 m, 20 m and 60 m resolution bands.

**Table 1.** Name and central wavelength of the Sentinel-2 spectral bands [1].

| Band | Central Wavelength (nm) | Spatial Resolution (m) |
| --- | --- | --- |
| 1—Coastal aerosol | 442.7 | 60 |
| 2—Blue | 492.4 | 10 |
| 3—Green | 559.8 | 10 |
| 4—Red | 664.6 | 10 |
| 5—Vegetation red edge | 704.1 | 20 |
| 6—Vegetation red edge | 740.5 | 20 |
| 7—Vegetation red edge | 782.8 | 20 |
| 8—NIR | 832.8 | 10 |
| 8A—Narrow NIR | 864.7 | 20 |
| 9—Water vapour | 945.1 | 60 |
| 10—SWIR—Cirrus | 1373.5 | 60 |
| 11—SWIR | 1613.7 | 20 |
| 12—SWIR | 2202.4 | 20 |

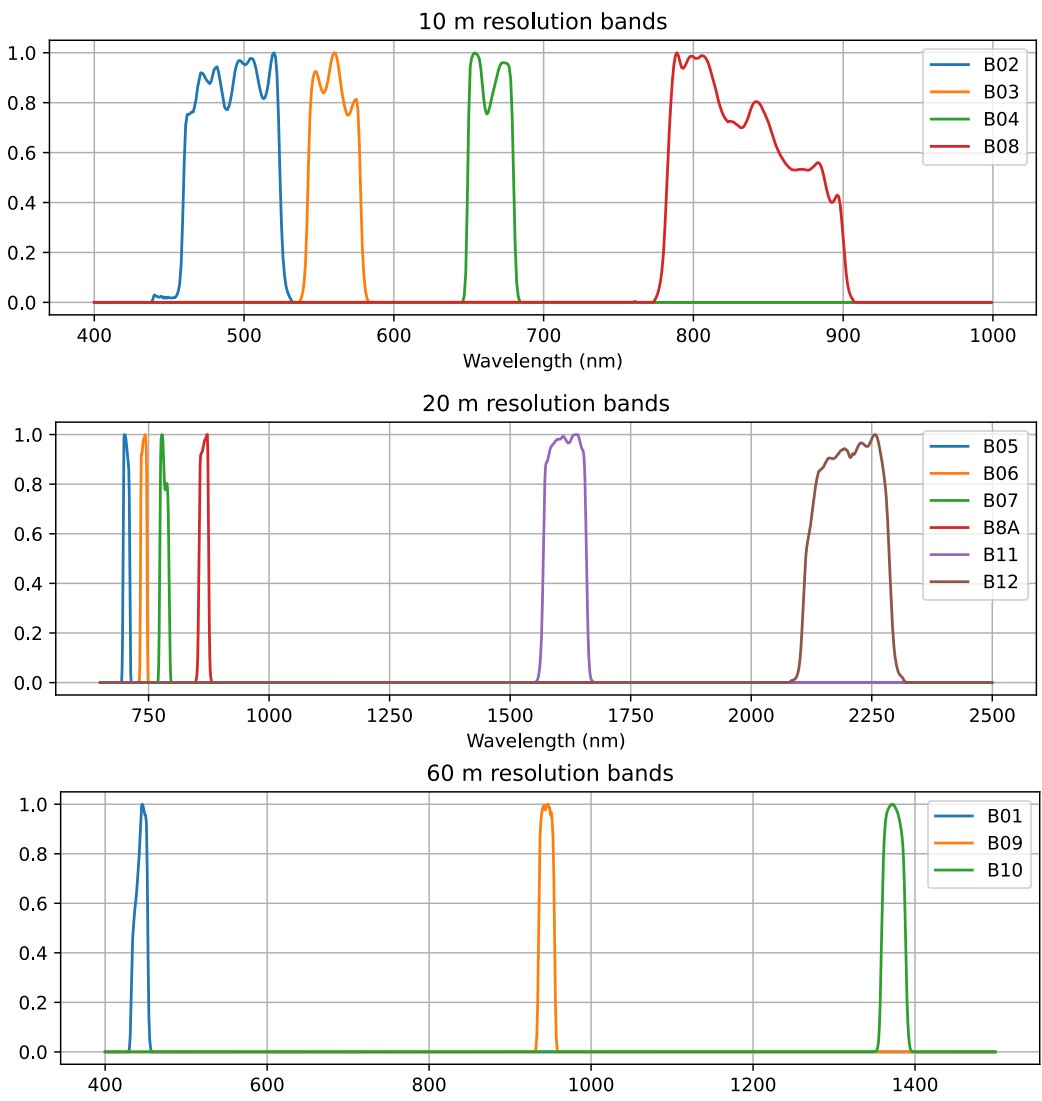

**Figure 1.** Sentinel-2A relative spectral responses from https://sentinels.copernicus.eu/documents/247904/685211/S2-SRF_COPE-GSEG-EOPG-TN-15-0007_3.0.xlsx, accessed on 10 May 2022.

### 1.2. S2 Radiometric Requirements

The Sentinel-2 Mission Requirements Document (MRD) [2] states that for the applications covered by this mission, the radiometric accuracy at top of atmosphere (TOA) has to be not worse than 3% (goal) to 5% (threshold). For inter-band radiometric calibration, 3% accuracy is also required.

These requirements allow the definition of error bounds for the band reconstruction tasks that we assess in this work. For TOA reflectances, we can aim for the 3% reconstruction error. In terms of surface reflectance, the accuracy of the MACCS-ATCOR Joint Algorithm (MAJA) processor [3,4] is 0.01 (not in %, but in reflectance values), and we can use this value as the requirement.

Given the fact that there are other errors in the measure (geometric registration between bands, Modulation Transfer Function (MTF) differences, etc.), achieving these error bounds can be considered rather ambitious.

Other approaches to define the reconstruction requirements could be used. For instance, [5] presents a radiometric uncertainty tool which allows estimation of the radiometric uncertainty associated with each pixel of a Sentinel-2 image in the TOA images provided by ESA. The approach integrates all the errors from the TOA reflectance to the L1C product, and typical values are greater than 10% for open sea, 5% to 15% in rice fields covered by water and 2% to 4% for land areas. We see that the 3% specification is very strict.

### 1.3. Directional Effects

Since the reflectance of surfaces depends on the observation and illumination directions [6], particular attention has to be payed to the acquisition geometry. Directional effects are especially important in (nearly) specular reflections, but also in the case of shadow or volume effects.

The MultiSpectral Instrument (MSI) is composed of two focal planes covering the VNIR and the SWIR channels, respectively, each one having an array of 12 detectors. Due to the shifted positioning of the detectors along the track direction on the focal planes, angular differences between the two alternating odd and even clusters of detectors are induced in the measurements. The parallax base/height (B/H) ratio ranges between 0.022 and 0.059. A similar issue occurs between the VNIR and SWIR detectors, resulting in an inter-band B/H which is less than 0.01 for the VNIR channels and less than 0.018 for the SWIR.

The values of the solar and sensor angles on a 5 km grid are provided in the L1C product metadata. We leverage this information in the band reconstruction algorithms that will be used in this work.

### 1.4. Specific Uses of S2 Bands

The spectral bands of Sentinel-2 allow the computation of a large variety of spectral indices other than NDVI that are useful for different applications. Table 2 presents a selection of several of them.

The RE bands have been proposed for chlorophyll estimation, burn severity assessment [7], LAI estimation [8] and non photosynthetic vegetation [9]. The SWIR bands have been proposed for dry mass vegetation [10] and water or moisture indices [11].

Although a thorough review of the literature is out of the scope of this paper, a bibliometric analysis shows that very few papers published after the launch of Sentinel-2 make an explicit use of the spectral particularities (RE and SWIR bands). Furthermore, a recent review about phenology monitoring using Sentinel-2 [12] shows that only one out of four published papers uses spectral information other than NDVI.

Some studies, for instance [13], claim that RE and SWIR bands during vegetation senescence appear to be important for machine learning-based classification. The concept of importance has to be nuanced, since it measures the errors made when the reflectance of those bands are replaced by random values. In order to have an accurate assessment of the usefulness of those variables, the classifiers should be retrained without them. On the

other hand, the same work shows that PSRI, which is computed from red, green and NIR, is also *important*, which may indicate a high correlation (and therefore redundancy) with RE bands.

**Table 2.** Spectral indices leveraging Sentinel-2 spectral bands for applications related to vegetation and water surfaces.

| Index | Formula | Application | Reference |
|---|---|---|---|
| $CI_{red-edge}$ | $\left(\frac{B7}{B5}\right) - 1$ | Chlorophyll, burnt areas | [7] |
| $CI_{green}$ | $\left(\frac{B7}{B3}\right) - 1$ | " | " |
| $REP$ | $705 + 35\frac{\frac{(B4+B7)}{2} - B5}{B6 - B5}$ | " | " |
| $MTCI$ | $\frac{B6 - B5}{B5 - B4}$ | " | " |
| $NDRE1$ | $\frac{B6 - B5}{B6 + B5}$ | " | " |
| $NDRE2$ | $\frac{B7 - B5}{B7 + B5}$ | " | " |
| $TRBI$ | $\frac{B12 + B6}{B8A}$ | LAI estimation | [8] |
| $NSSI$ | $\frac{B8A - B7}{B8A + B7}$ | Non-photosynthetic vegetation | [9] |
| $PSRI$ | $\frac{B4 - B2}{B6}$ | Senescent vegetation | [14] |
| $STI$ | $\frac{B11}{B12}$ | Tillage, dry vegetation | [10] |
| $NDWI$ | $\frac{B3 - B8A}{B3 + B8A}$ | Water bodies | [15] |
| $NDWI$ | $\frac{B8 - B11}{B8 + B11}$ | " | [11] |
| $NDWI$ | $\frac{B8 - B12}{B8 + B12}$ | " | " |

Another work supporting the interest of RE and SWIR bands is [16], where they are shown to be useful for gross primary productivity estimation in grasslands. Using regression approaches, the authors show that those bands are useful to predict the target variable, but do not study whether by using more complex regressors the error without those bands could be reduced.

It is interesting to note that other works, for instance [17], show that NDVI is best suited to monitor grass phenology rather than more sophisticated VIs using RE and SWIR bands. Another example is [18], where it is shown that the RE bands of Sentinel-2 do not perform well for the estimation of chlorophyll content changes in certain crops. One should note that before the launch of Sentinel-2 the same community had great expectations for these bands for the same application [19]. However, at the time, the authors already suggested that using the green band in $CI_{green}$ also seemed very promising and therefore further research was required.

The apparent contradictions between these different works are likely due to the fact that different experimental settings, different data and different applications were involved.

Further, we find that works on the usefulness of spectral bands are usually addressed only from the point of view of demonstrating that a particular phenomenon has a signature in a particular band. For instance, a recent publication [20] proposed additional SWIR bands in order to detect non-photosynthetic vegetation and crop residues. The study indeed shows that these objects cannot be detected with the SWIR bands of Landsat-8. However the cited work does not analyze how the complete set of Landsat-8 bands could be used to retrieve a signature of the phenomenon at hand.

As of this writing and to the best of our knowledge, the most thorough study of the *usefulness* of Sentinel-2's spectral bands is [21]. This reference is actually a detailed literature review of the use of hyperspectral imagery with the goal of proposing synergies with Sentinel-2 in order to overcome the limitations of space-borne hyperspectral sensors (spatial resolution, revisit time and signal-to-noise ratio). Interestingly, the review shows how the current set of Sentinel-2 bands constitutes in itself a very wise choice for many applications. However, the limit of such a meta-analysis is that there cannot be a holistic view of the problem, since the pertinence of each spectral range is performed in isolation in the reviewed literature. Indeed, this prevents discovering redundancies between different

bands. For instance, this reference excludes uses for geological and lithological mapping, such as [22–24], where the higher resolution of Sentinel-2's NIR bands is assessed for the estimation of iron oxides.

We think this supports the idea of performing a purely data-driven approach over a large dataset and with an application-agnostic point of view. However, the work presented in this paper is just a modest demonstration of what could be done by exploiting the existing Sentinel-2 archive.

Finally, we will stress again the fact that we do not claim that some Sentinel-2 bands do not contain useful information. We want to assess the possibility of reconstructing this information by leveraging redundancies among the complete set of spectral bands. This reconstruction will, of course, contain errors, and the goal here is to give bounds allowing informed design trade-offs for future systems.

## 2. Materials and Methods

### 2.1. Data Preparation

For this study, a set of 128 Sentinel-2 tiles was used. Figure 2 illustrates the geographic distribution of these tiles. For each tile, a single date was used and the selection was random for the period from early 2016 to the end of 2020. The goal was to cover a wide range of geographic areas and seasons. For each acquisition, the data were obtained at two processing levels: 1C (from PEPS, CNES mirror of Sentinel data) and 2A (from Theia's catalog ), the latter having been produced by the MAJA processor. This allows us to use accurate masks at the pixel level for clouds, cloud shadows and saturation effects.

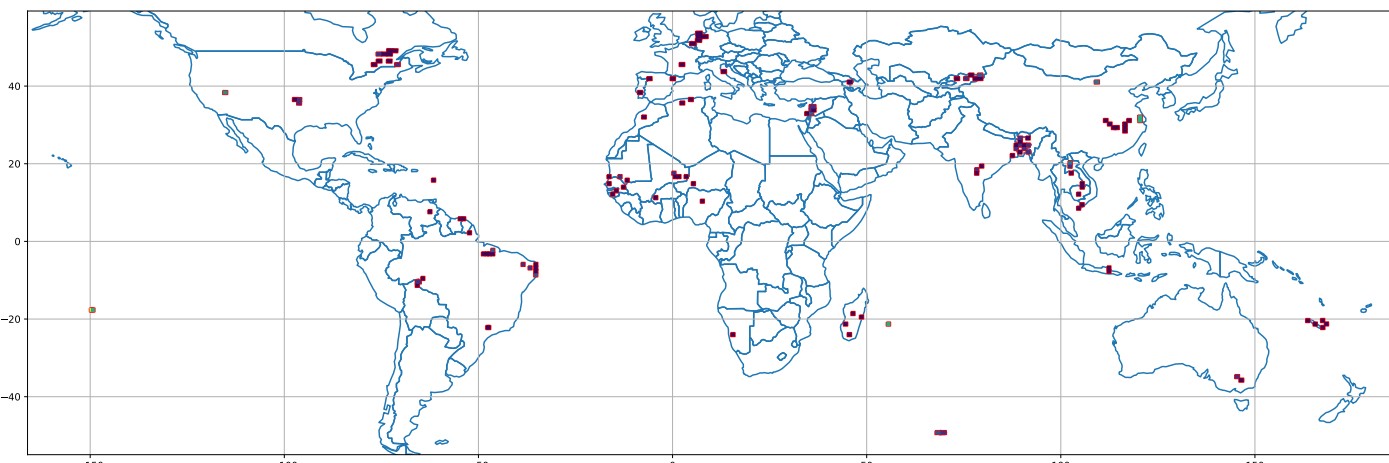

**Figure 2.** Geographic distribution of the tiles used for the study.

For each acquisition, 100,000 pixels were sampled. Only non-saturated pixels were selected, regardless of their cloud or shadow status. Pixel positions were selected on the 20 m resolution grid. For each 20 m pixel position, the following information was recorded:

- whether the pixel was detected as a cloud or a shadow (without distinction between these two states),
- the reflectance in the 20 m bands for levels 1C and 2A,
- the reflectance of the four corresponding pixels of each of the 10 m resolution bands for levels 1C and 2A,
- the reflectance at the 20 m pixel position of the 60 m resolution bands after bicubic resampling for level 1C,
- the solar and viewing angles for each pixel.

For the analyses performed in the following sections, we split the data at the tile level. This means that all the pixels used for testing (measures of accuracy of the reconstructions) belong to tiles for which no pixel was used for training or even intermediate validation.

In the experiments carried out in this work, we randomly select 100 tiles and do a 80/20% split at the tile level for training and testing purposes. This means that training and testing pixels come from different tiles and dates. The training set is further split into proper training samples (80%) and validation samples (20%), the latter being used for monitoring the convergence of the training. For each experiment (i.e., set of predicted bands and set of predictor bands), the experiment is repeated 10 times by selecting a different set of 100 tiles among the 128 available. This allows checking for possible selection biases and allows further assessment of the robustness of the regressions.

Further, only clear pixels (non-cloudy and not shadow) are used for training and validating models. This reduces the number of available pixels. On average, each experiment uses $3.86928 \times 10^6$ training samples, 967,320 validation samples and $1.2582 \times 10^6$ testing samples and is repeated 10 times.

The dataset has been made public [25] and is available for other researchers to reproduce and improve the work presented in this paper.

### 2.2. Regression Model

As stated in the introduction, we aim at estimating a subset of the Sentinel-2 bands from the other ones. This estimation will be done using regression techniques. The regression algorithms will be calibrated and validated using the data described in Section 2.1. In this section, we describe the regression approach chosen.

### 2.2.1. Reflectance Estimation with Associated Uncertainties

The regression problem is posed as the estimation of one or several spectral bands as a nonlinear combination of a disjointed set of the available bands. For the prediction of a single-band, this can be written as:

$$\widehat{\rho_i} = f(\{\rho_{j \neq i}\}, \vec{\theta}),$$

that is, prediction of reflectance of band *i* is a function of the measured reflectances of the other bands and a set of parameters $\vec{\theta}$ containing other pertinent information, such as solar and sensor angles. The regression can jointly estimate several spectral bands in a set *I*:

$$\{\widehat{\rho_i}\}_{i \in I} = f(\{\rho_j\}_{j \notin I}, \vec{\theta}) \tag{1}$$

The regression procedure should also produce a credibility interval of the estimation of the target variable. (We use the term *credibility interval* instead of *confidence interval* because we adopt a Bayesian point of view: we consider the estimated value is a random variable and the bounds of the interval are fixed, while the use of confidence intervals considers the bounds as random variables that result from repeated measures.) In order to do this, instead of regressing over the expected mean, we can implement a regression of the mean and the variance of the target variable. Estimating a mean and a variance means that we are assuming a Gaussian error model.

At inference (estimation) time, the mean will be used for variable estimation (in a Gaussian model the mean is the value with the highest probability), and the variance will be used to give the credibility interval.

Given a target value $y$ (in our case $\rho_i$) and the estimates $\hat{\mu}$ and $\hat{\sigma}$, the predictive likelihood of the target value given the estimates is the Gaussian distribution whose probability density function is

$$p(y|\hat{\mu}, \hat{\sigma}) = \frac{1}{\sqrt{2\pi\hat{\sigma}^2}} e^{\frac{(y-\hat{\mu})^2}{2\hat{\sigma}^2}}$$

We can therefore pose the regression problem as the minimization of a cost function given by the negative log-likelihood [26]. The log-likelihood takes the form:

$$log(p(y|\hat{\mu}, \hat{\sigma})) = -\frac{1}{2}\log(2\pi) - \frac{1}{2}\log\hat{\sigma}^2 - \frac{(y - \hat{\mu})^2}{2\hat{\sigma}^2}$$

Therefore, after removing the constant term and a multiplicative factor, the cost function to be minimized is:

$$\mathcal{L} = \sum_i \log\hat{\sigma}_i^2 + \frac{(y_i - \hat{\mu}_i)^2}{\hat{\sigma}_i^2}$$

where the sum is taken over the training samples.

Beyond being the correct theoretical loss under a Gaussian error model, this penalty function can be interpreted as follows :

- the term $(y_i - \hat{\mu}_i)^2$ penalizes the errors between the target value and the estimated mean;
- these errors are weighted by the uncertainty estimation: larger errors will need larger values of $\sigma_i$ to lower the penalty;
- in order to avoid allowing large errors on $\mu_i$ by always estimating large values of $\sigma_i$, large values of $\sigma_i$ are also penalized by the first term in the loss.

### 2.2.2. Regression Algorithm

The regression algorithm will have to find the approximation of function $f$ in Equation (1) that minimizes the cost function described above. Since we don't have prior knowledge of the shape of $f$, we choose to use a non-parametric approach. Among non-parametric algorithms for regression, feed-forward neural networks (multi-layer perceptrons, MLPs) seem a good choice because they are universal function approximators [27] that can be used in a multi-variate input and output setting and with custom cost functions. Conversely, other choices have limitations. For instance, linear and logistic regressions impose a strong prior on the shape of $f$, and random forest regression cannot predict several targets. The main drawback of neural networks is their lack of interpretability.

MLPs are composed of fully connected linear layers (sets of neurons computing linear combinations of the inputs) followed by nonlinearities $\phi$ called activation functions. Figure 3 illustrates an MLP with a single hidden layer with five neurons. A large number of layers with different numbers of neurons can be used. Training such a network consists in finding the set of weights $w_i$ that minimize the loss function for the set of training samples. Optimization is performed by stochastic gradient descent.

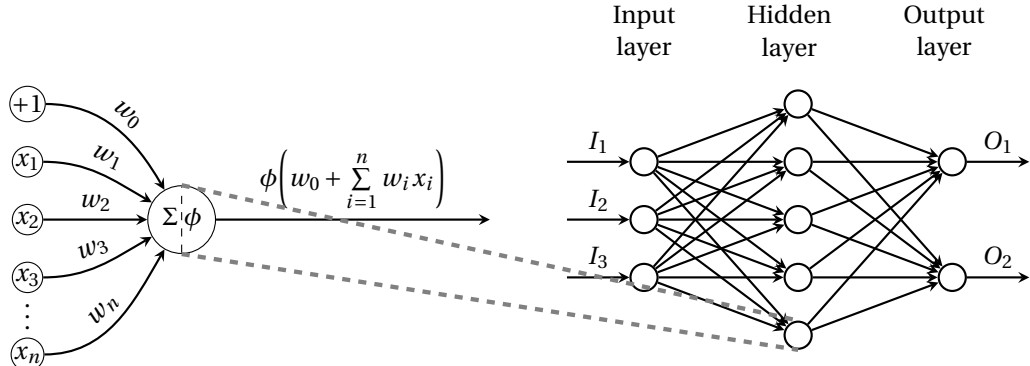

**Figure 3.** A multi-layer perceptron with one input layer, one hidden layer and one output layer. Diagram adapted from https://github.com/PetarV-/TikZ, accessed on 10 May 2022.

Another interesting property of MLPs is that they can be combined as elementary bricks in more complex architectures. This allows the introduction of some structure in

the processing, which brings interpretability and the possibility of introducing some prior knowledge. We will develop this point in the next section.

### 2.2.3. Network Architecture

As stated above, the regression neural network will estimate the reflectances of the target bands using the reflectances of the other bands as predictors. All computations are performed for individual pixels. In order to take into account BRDF effects, the solar and sensor angles (both azimuth and zenith, as described in Section 1.3) are also used as predictors. More precisely, the sines and cosines of each angle are used.

Instead of using all predictors (reflectances and angles) together in a flat vector as input for an MLP as in Figure 3, we use an attention mechanism where the angular information modulates the spectral values. This is implemented as illustrated in Figure 4. First, the spectral and angular information are fed to the *Angular MLP* which is used to generate an attention mask. An attention mask is a vector of real numbers in $[0, 1]$ with the same number of components as the data on which the attention is being applied. In our case, this is the vector of spectral bands. The *Angular MLP* is a standard MLP with a single hidden layer containing eight neurons and a *SoftMax* layer as output. The *SoftMax* function is an exponential normalization that maps a set of values to the unit interval (simplex in more than one dimension) $\sigma : \mathbb{R}^K \to [0, 1]^K$ and is defined by:

$$\sigma(\mathbf{z})_i = \frac{e^{z_i}}{\sum_{j=1}^{K} e^{z_j}} \quad \text{for } i = 1, \dots, K \text{ and } \mathbf{z} = (z_1, \dots, z_K) \in \mathbb{R}^K,$$

where $z_i$ are the outputs of the layer preceding the *SoftMax*.

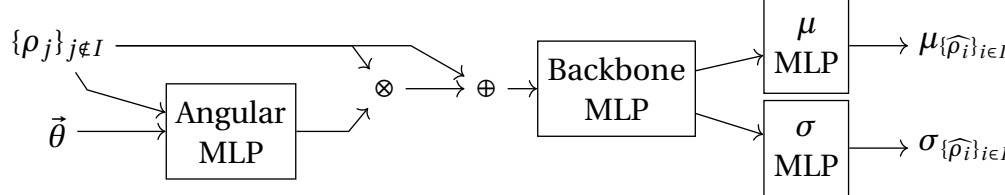

**Figure 4.** Overview of the nonlinear regression of a set of spectral bands using other bands and angular information as predictors assuming a Gaussian error model.

Therefore, the *Angular MLP* learns a set of multiplicative weights (this operation is represented by the $\otimes$ symbol in Figure 4) that will be applied to the input reflectances in order to perform an angular correction. It is interesting to note that this angular correction takes into account the spectral information itself, that is, the reflectances and the angles are both used to estimate the correction. It is therefore a kind of self-attention mechanism [28].

A residual connection (a simple, elementwise addition, represented by $\oplus$ in Figure 4) is used after the attention mask in order to keep spectral information that could be excessively removed by the attention mechanism before entering the *Backbone MLP*. The latter is used to embed the predictors into a feature space that will be used to feed the two modules used to estimate the target values and their uncertainties, respectively.

The backbone part (a three-hidden-layer MLP with 10 neurons per layer) allows modeling of the correlation between the target variables and their uncertainties. The independent MLP branches (with the same architecture as the backbone) for $\mu$ and $\sigma$ get specialized into the estimation of each set of information. Performing the regression for several target variables with the same network is a kind of multitask learning that is able to leverage the correlation between target variables and is more efficient than preforming single target regressions.

For numerical stability and positivity constraints, instead of estimating the $\sigma$ or $\sigma^2$, we estimate $\log \sigma$.

The output activation functions for the mean and the variance estimations are hyperbolic tangents so that the values are contained in the $[-1, 1]$ interval. The output value

is then rescaled into a predefined interval, $[-0.2, 1.3]$ for $\mu$ and $[1 \times 10^{-5}, 1.5]$ for $\sigma^2$. The rescaling for $\mu$ allows taking into account the fact that L2A reflectances can sometimes be negative due to over-correction. Reflectances can also be higher than one in specular conditions. The rescaling intervals could be learned from the data, but we set them for simplicity.

The regression of several bands simultaneously is done by a straightforward extension of the single target case. The output layers for both the means and the variances will have as many neurons as target variables. The loss function is just the sum of the losses for each target variable.

The network is trained for 100 epochs using an Adam optimizer [29] with a learning rate of 0.001 and a batch size of 256.

### 2.3. Measuring Redundancies in Sentinel-2 Bands

To assess the quality of the spectral regression approaches, we will analyze the statistical dependence between all the pairs of Sentinel-2 bands. Instead of measuring correlations, which are limited to linear (Pearson correlation) or monotonic (Spearman correlation) dependencies, we will use the mutual information, $I$. It measures dissimilarity between the joint distribution of a pair of variables and the product of the marginals. It is therefore a measure of the distance to general statistical independence:

$$I(X;Y) = D_{KL}(P_{(X,Y)} \| P_X \otimes P_Y),$$

where $D_{KL}$ is the Kullback–Leibler divergence. The mutual information can also be written in terms of entropies ($H$) as follows:

$$I(X;Y) = H(X,Y) - H(X|Y) - H(Y|X) = H(Y) - H(Y|X) = H(X) - H(X|Y),$$

and it is therefore a measure of the amount of uncertainty about one variable once the other is known. The mutual information is positive, but it is not upper-bounded. Therefore, we use a normalized version using the entropies of each variable:

$$I_{norm}(X;Y) = \frac{I(X;Y)}{\sqrt{H(X)H(Y)}}$$

We will study this measure for both L1C and L2A data.

## 3. Results

### 3.1. Redundancies in Sentinel-2 Bands

As stated in Section 2.3, we start by analyzing the redundancies in Sentinel-2 spectral bands. Figure 5 displays the values of the normalized mutual information correlation for all pairs of bands of L1C (left) and L2A (right) data.

Both levels of processing show the same patterns and nearly the same values, although L2A has slightly lower values of dependence. This may indicate that atmospheric corrections are able to remove effects with high correlation across bands.

We observe high values for the red edge bands between B5 and the red band, and between the two SWIR bands. Interestingly, B5 presents a relatively low dependence with respect to B6 and B7, and there is very small redundancy between B8 and B8A (it is, for instance, lower than between green and B5).

The highest values of mutual information are obtained between adjacent bands of the B6, B7, B8A triplet; B7 being the most similar to the others. B7 therefore seems a good candidate for reconstruction from other bands.

One limitation of this analysis is that only pairs of bands are compared, and therefore it is impossible to assess if the redundancies between, for instance, B7 and B6 are complementary to those between B7 and B8A, which would allow better reconstruction of B7 from the other two than if these redundancies were the same.

It is also interesting to note that B5 has all values higher than 0.4 (except for B8), which may indicate either the possibility of reconstructing it from the other bands, or conversely, of it being some sort of pivotal band to reconstruct the others.

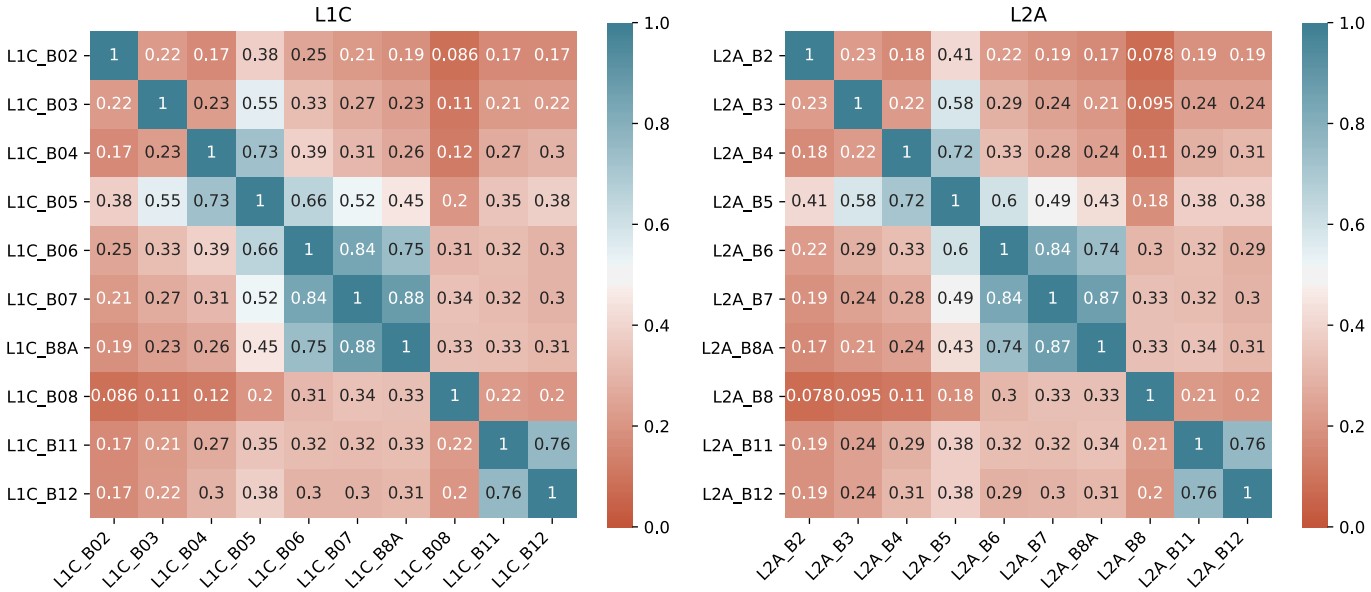

**Figure 5.** Normalized mutual information.

The relatively low value of the mutual information between B8 and B8A may seem surprising since the latter is a subset of the former. Actually, this value is the same for B7 and B8, which are adjacent (see Figure 6). However, B8A has a width less than 20% of that of B8.

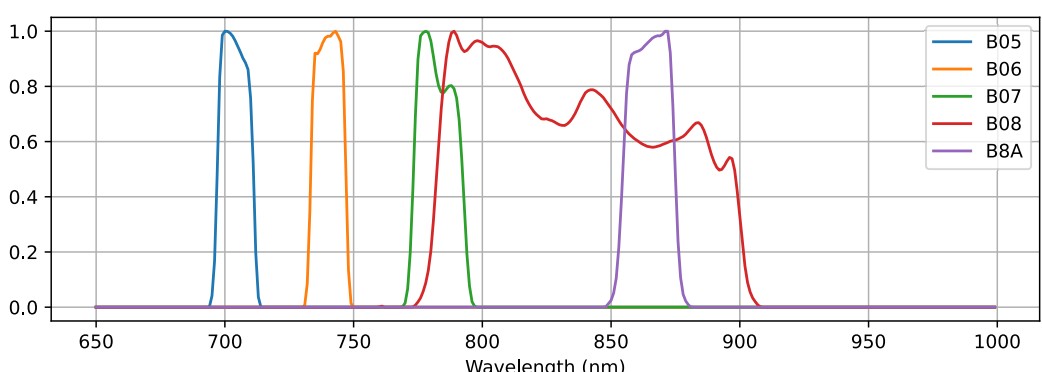

**Figure 6.** Red edge and NIR bands.

This means that these measures of mutual information are lower bounds of the amount of information that could be reconstructed from other bands.

### 3.2. Single Band Regression

We present in this section the performance of the reconstruction of each spectral band by applying the neural network regression algorithm described in Section 2.2. As stated before, only the 20 m bands are reconstructed, and the following data are used as predictors:

- the sines and cosines of the four observation angles,
- all 20 m bands except for the target one,
- the values of the four 10 m pixels for B2, B3, B4 and B8 associated with the 20 m target pixel,

- and, only for L1C, the value of the three 60 m bands interpolated (with a bicubic interpolator) to the coordinate of the center of the 20 m pixel.

Each regression case is repeated 10 times using the protocol described at the end of Section 2.1.

### 3.2.1. Analysis of Errors

Validation metrics are computed across all experiments and reported in Tables 3 and 4 for L1C and L2A data, respectively. The tables present the root mean square error (RMSE), the mean absolute error (MAE), the relative error (RE) and the coefficient of determination ($R^2$). The rows of the tables are sorted by increasing values of RE for L1C and RMSE for L2A.

**Table 3.** Single-band regression results for L1C. The colors in the RE (relative error) column indicate whether the specification is fulfilled (light gray), nearly fulfilled (middle gray) or unfulfilled (dark gray).

| Band | RMSE | MAE | RE | $R^2$ |
|------|------|-----|-----|-------|
| B07 | $7.17 \times 10^{-3}$ | $3.90 \times 10^{-3}$ | $2.96 \times 10^{-2}$ | $9.96 \times 10^{-1}$ |
| B06 | $1.82 \times 10^{-2}$ | $4.77 \times 10^{-3}$ | $3.61 \times 10^{-2}$ | $9.88 \times 10^{-1}$ |
| B8A | $1.57 \times 10^{-2}$ | $5.33 \times 10^{-3}$ | $3.69 \times 10^{-2}$ | $9.91 \times 10^{-1}$ |
| B05 | $1.57 \times 10^{-2}$ | $4.46 \times 10^{-3}$ | $3.79 \times 10^{-2}$ | $9.92 \times 10^{-1}$ |
| B12 | $1.50 \times 10^{-2}$ | $9.18 \times 10^{-3}$ | $9.35 \times 10^{-2}$ | $9.83 \times 10^{-1}$ |
| B11 | $1.83 \times 10^{-2}$ | $1.26 \times 10^{-2}$ | $1.51 \times 10^{-1}$ | $9.85 \times 10^{-1}$ |

**Table 4.** Single-band regression results for L2A. The colors in the RMSE (relative error) column indicate whether the specification is fulfilled (light gray), nearly fulfilled (middle gray) or unfulfilled (dark gray).

| Band | RMSE | MAE | RE | $R^2$ |
|------|------|-----|-----|-------|
| B5 | $7.33 \times 10^{-3}$ | $4.96 \times 10^{-3}$ | $2.07 \times 10^{-1}$ | $9.95 \times 10^{-1}$ |
| B6 | $8.26 \times 10^{-3}$ | $5.04 \times 10^{-3}$ | $1.28 \times 10^{-1}$ | $9.96 \times 10^{-1}$ |
| B7 | $8.42 \times 10^{-3}$ | $5.02 \times 10^{-3}$ | $1.18 \times 10^{-1}$ | $9.97 \times 10^{-1}$ |
| B8A | $1.11 \times 10^{-2}$ | $6.14 \times 10^{-3}$ | $2.23 \times 10^{-1}$ | $9.95 \times 10^{-1}$ |
| B12 | $1.49 \times 10^{-2}$ | $9.49 \times 10^{-3}$ | $2.41 \times 10^{-1}$ | $9.75 \times 10^{-1}$ |
| B11 | $2.06 \times 10^{-2}$ | $1.36 \times 10^{-2}$ | $4.05 \times 10^{-1}$ | $9.81 \times 10^{-1}$ |

In Section 1.2, we concluded that 3% error for L1C and 0.01 in surface reflectance values for L2A were good targets for band reconstruction. Of course, we are measuring reconstruction errors using data which itself may have errors, even if they are below the radiometric specifications. Therefore, the error bounds need not to be taken very strictly. Finally, Sentinel-2 can be considered to be over-specified in terms of radiometric quality for most applications, which makes using these error bands rather conservative from our point of view.

We see that for L1C only the reconstruction of B7 has an RE lower than 3%, although the other red edge and NIR bands are below 3.8%. For L2A, B5, B6 and B7 have an RMSE lower than 0.01, and B8A is only slightly above this level.

Estimating the noise in surface reflectances using the RMSE can suffer from strong outliers. The MAE gives a measure that is robust to these cases and shows that even B12 could be considered for reconstruction.

The error values presented in Tables 3 and 4 are averages over the validation samples and do not show the proportion of pixels that do not fulfill the radiometric requirements. For this purpose, Tables 5 and 6 show the percentage of pixels whose error is lower than a given threshold.

Table 5 presents, for each L2A band, the percentage of pixels whose error is larger than a given threshold (from 0.01, which is the accuracy of the L2A processor, up to 0.025). We see that even for the best-predicted bands (in the red edge), less than 90% of the pixels fulfill the requirements. However, lowering the requirement accuracy to 0.015, 95% compliance is achieved for these three bands.

**Table 5.** Percentage of pixels beyond a given absolute error threshold (L2A).

| Band | 0.01 | 0.015 | 0.02 | 0.025 |
|------|------|-------|------|-------|
| 9 | 4.76 | 1.94 | 0.91 | |
| B6 | 12.60 | 5.35 | 2.67 | 1.51 |
| B7 | 12.47 | 5.31 | 2.60 | 1.39 |
| B8A | 19.19 | 8.73 | 4.40 | 2.47 |
| B11 | 46.75 | 34.45 | 25.03 | 18.11 |
| B12 | 33.81 | 22.00 | 14.96 | 10.41 |

Table 6 shows the same results for L1C data. The performance seem to be much better than for L2A, but we must remember that the requirements for L1C are given as relative errors (the error must not exceed 3%).

**Table 6.** Percentage of pixels beyond a given absolute error threshold (L1C).

| Band | 0.01 | 0.015 | 0.02 | 0.025 |
|------|------|-------|------|-------|
| B05 | 6.86 | 2.37 | 1.21 | 0.78 |
| B06 | 8.09 | 3.07 | 1.42 | 0.79 |
| B07 | 8.95 | 3.15 | 1.19 | 0.51 |
| B8A | 13.49 | 4.83 | 1.87 | 0.82 |
| B11 | 43.70 | 30.51 | 20.97 | 14.35 |
| B12 | 30.20 | 20.38 | 14.14 | 9.75 |

Table 7 shows the percentage of validation pixels compliant with different error thresholds. We see that the requirement has to be lowered from 3% to 10% in order to get 95% compliance for the red edge and NIR bands. This poor performance is mainly due to high relative errors in the low reflectances. Tables 8–12 show the compliance with relative error thresholds for different intervals of reflectances. The results confirm that reflectances lower than 0.1 contain most of the errors.

**Table 7.** Percentage of pixels beyond a given relative error threshold (L1C).

| Band | 0.03 | 0.05 | 0.1 |
|------|------|------|-----|
| B05 | 37.23 | 20.44 | 5.34 |
| B06 | 26.50 | 10.74 | 2.36 |
| B07 | 21.99 | 8.97 | 2.10 |
| B8A | 30.11 | 13.16 | 3.66 |
| B11 | 69.33 | 51.53 | 22.93 |
| B12 | 73.79 | 58.25 | 30.10 |

**Table 8.** Percentage of pixels beyond a given relative error threshold for reflectances in [0, 0.1] (L1C).

| Band | 0.03 | 0.05 | 0.1 |
|------|------|------|-----|
| B05 | 49.66 | 29.34 | 7.85 |
| B06 | 39.48 | 16.89 | 2.25 |
| B07 | 45.16 | 23.58 | 4.94 |
| B8A | 48.46 | 27.90 | 7.02 |
| B11 | 75.16 | 60.25 | 32.95 |
| B12 | 74.92 | 60.26 | 33.89 |

**Table 9.** Percentage of pixels beyond a given relative error threshold for reflectances in [0.1, 0.25] (L1C).

| Band | 0.03 | 0.05 | 0.1 |
|------|------|------|-----|
| B05 | 35.01 | 17.53 | 3.58 |
| B06 | 25.18 | 8.93 | 1.04 |
| B07 | 21.63 | 8.11 | 1.05 |
| B8A | 30.70 | 12.17 | 1.69 |
| B11 | 69.40 | 51.48 | 20.39 |
| B12 | 73.14 | 57.22 | 27.93 |

**Table 10.** Percentage of pixels beyond a given relative error threshold for reflectances in [0.25, 0.5] (L1C).

| Band | 0.03 | 0.05 | 0.1 |
|------|------|------|-----|
| B05 | 7.98 | 2.69 | 0.37 |
| B06 | 16.71 | 5.98 | 0.42 |
| B07 | 14.57 | 4.03 | 0.26 |
| B8A | 19.82 | 4.63 | 0.24 |
| B11 | 65.31 | 45.43 | 15.98 |
| B12 | 70.94 | 52.66 | 19.58 |

**Table 11.** Percentage of pixels beyond a given relative error threshold for reflectances in [0.5, 0.75] (L1C).

| Band | 0.03 | 0.05 | 0.1 |
|------|------|------|-----|
| B05 | 10.06 | 3.15 | 0.29 |
| B06 | 10.20 | 2.93 | 0.22 |
| B07 | 17.32 | 4.88 | 1.13 |
| B8A | 20.40 | 5.26 | 0.25 |
| B11 | 53.24 | 27.82 | 3.57 |
| B12 | 52.10 | 31.80 | 5.49 |

**Table 12.** Percentage of pixels beyond a given relative error threshold for reflectances in [0.75, 1] (L1C).

| Band | 0.03 | 0.05 | 0.1 |
|------|------|------|-----|
| B05 | 13.94 | 8.35 | 1.36 |
| B06 | 5.30 | 1.17 | 0.07 |
| B07 | 3.87 | 1.30 | 0.73 |
| B8A | 6.61 | 0.86 | 0.02 |
| B11 | 85.90 | 75.64 | 51.28 |
| B12 | 92.86 | 90.00 | 81.43 |

Figures 7 and 8 display scatterplots of predicted versus real reflectance values for the L1C and L2A bands, respectively. For clarity in the visualization, these scatterplots are generated with a small random sample of the validation data. Nevertheless, they show the general behavior and are coherent with the metrics presented in the tables above.

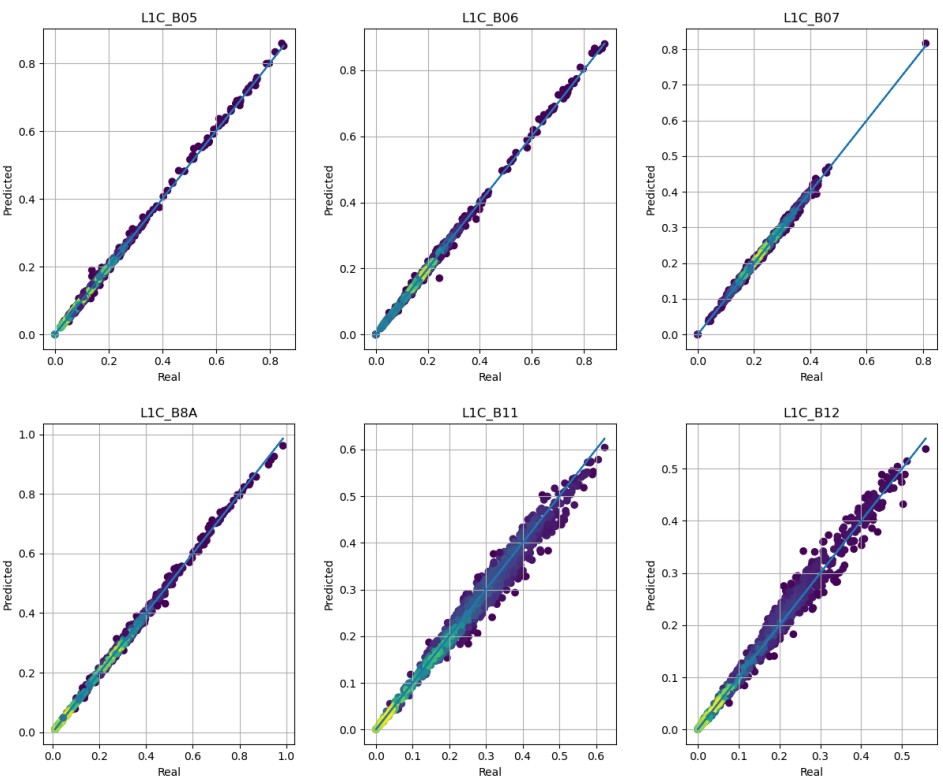

**Figure 7.** Scatterplots for the single-band regression (L1C). The colors ▬▬▬ indicate the density of points.

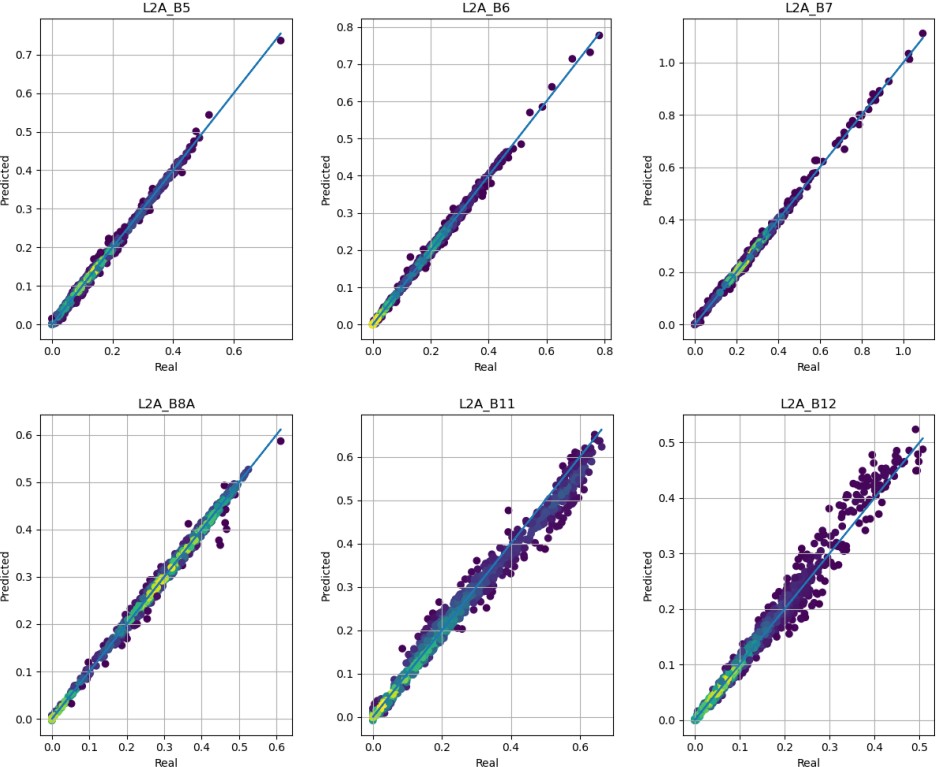

**Figure 8.** Scatterplots for the single-band regression (L2A). The colors ▬▬▬ indicate the increasing density of points.

To complete the analysis of the errors, we present the histograms of the errors (true minus predicted reflectance) using the complete validation dataset (about 5 million pixels). Figure 9 shows the histograms for the L1C bands and Figure 10 for the L2A bands.

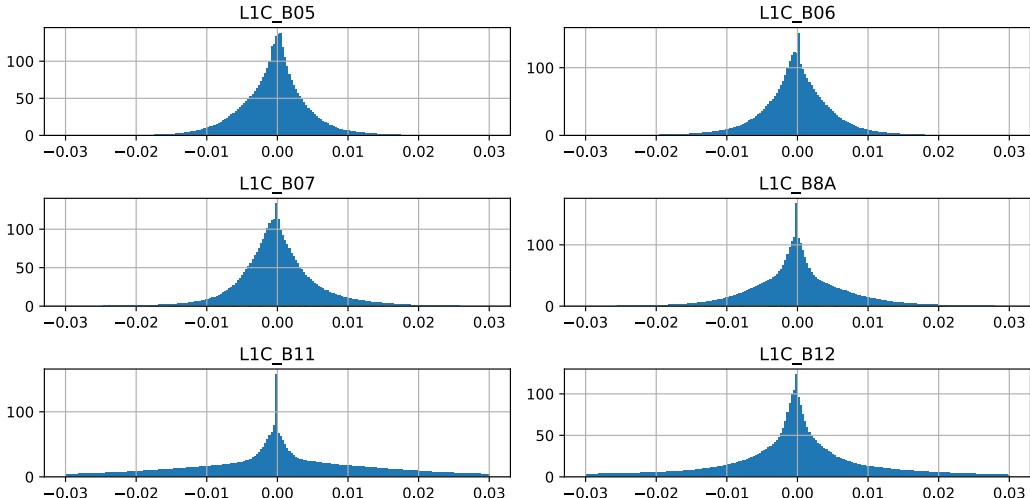

**Figure 9.** Histograms of the errors (true value minus prediction) for the L1C bands.

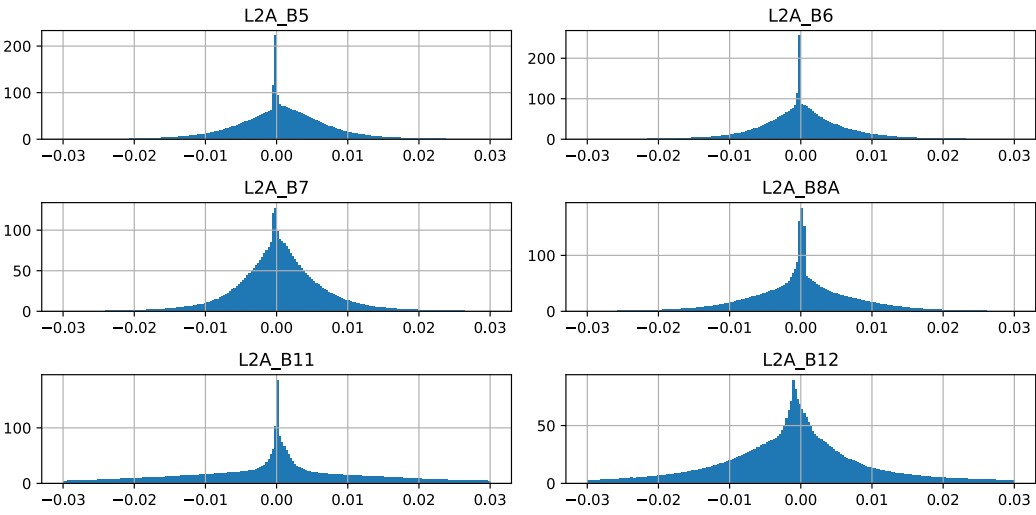

**Figure 10.** Histograms of the errors (true value minus prediction) for the L2A bands.

### 3.2.2. Analysis of the Uncertainty Estimation

As explained in Section 2.2.1, the regression model is also able to estimate the uncertainty of the predicted value by associating a variance with it. Since this variance is an estimation itself, its meaningfulness needs to be assessed.

The loss function used to train the model was chosen assuming a Gaussian error model. The histograms in Figures 9 and 10 show that the distributions of the errors are not Gaussian. However, these distributions are mono-modal, which may allow use of the estimated variance as a good proxy for the uncertainty of the estimation. In order to check this hypothesis, we will measure the proportion of pixels having errors higher than a given proportion of the variance.

In the case of a Gaussian distribution, we have $P(\mu - 1\sigma \leq X \leq \mu + 1\sigma) \approx 68.27\%$, $P(\mu - 2\sigma \leq X \leq \mu + 2\sigma) \approx 95.45\%$ and $P(\mu - 3\sigma \leq X \leq \mu + 3\sigma) \approx 99.73\%$.

We can therefore compute the proportion of pixels having an absolute error lower than $\sigma$, $2\sigma$ and $3\sigma$ and compare the results to the probability values above.

Tables 13 and 14 present the above-mentioned proportions of pixels whose errors are within the bounds given by the estimated sigma. We see that, although not identical, the proportions are relatively similar to what one should get in the Gaussian case.

**Table 13.** Probability of the absolute error being lower than $n \times \sigma$ (L1C).

| Band | $\sigma$ (68.27%) | $2\sigma$ (95.45%) | $3\sigma$ (99.73%) |
|------|------|------|------|
| B05 | 68.80 | 92.26 | 98.29 |
| B06 | 70.23 | 93.14 | 98.53 |
| B07 | 70.82 | 93.27 | 98.43 |
| B8A | 70.36 | 91.99 | 97.57 |
| B11 | 56.86 | 84.11 | 94.94 |
| B12 | 61.57 | 87.50 | 96.23 |

**Table 14.** Probability of the absolute error being lower than $n \times \sigma$ (L2A).

| Band | $\sigma$ (68.27%) | $2\sigma$ (95.45%) | $3\sigma$ (99.73%) |
|------|------|------|------|
| B5 | 65.68 | 90.00 | 96.87 |
| B6 | 71.84 | 93.85 | 98.70 |
| B7 | 74.28 | 94.50 | 98.69 |
| B8A | 70.07 | 92.13 | 97.93 |
| B11 | 56.18 | 81.51 | 93.04 |
| B12 | 64.70 | 91.15 | 97.91 |

It is important to understand that the value of $\sigma$ is provided by the regression algorithm as a prediction. These results show that this prediction of $\sigma$ is indeed a good proxy for the probability of the reflectance estimation being in the predicted interval. Therefore, the estimation of $\sigma$ can be a threshold and used as a validity mask for the estimations.

*3.3. Double-Band Regression*

We present here the results for the case where two bands are predicted from the others. This case will, of course, produce higher estimation errors because for each predicted band there is one fewer predictor.

Tables 15 and 16 present the results for the L1C and the L2A data. Each row in the tables presents the results for a pair of bands jointly predicted. The same quality metrics as for single-band regression are presented. Each table has 15 rows since we evaluate all possible combinations of pairs of bands.

The rows in Table 15 are sorted in increasing order of the maximum relative error of the pair of bands. This allows one to quickly see that only one pair of L1C bands (B06 and B07) can be predicted with less than 3% error, and that another pair (B07 and B8A) is slightly above this threshold.

For the L2A data presented in Table 16, the rows are sorted by increasing RMSE using the maximum of the pair in each row. In this case, only the pair (B5, B8A) fulfills the 0.01 error threshold, although the pair (B5, B6) is not much above this threshold.

Figure 11 presents the scatterplots for the two best pairs of L1C bands (the two first rows in Table 15. Altough the scatterplots are generated by subsampling the test dataset for readability, one can see that the estimations are unbiased and have a small dispersion around the regression lines. One can also see that part of the error comes from pixels with reflectances higher than one, for which there is underestimation. Since the regression algorithm is configured to yield reflectances in the $[-0.2, 1.3]$ interval, we can expect that the error in this interval is smaller than what is reported in the tables.

**Table 15.** Double-band regression results for L1C. The colors in the RE (relative error) columns indicate whether the specification is fulfilled (light gray), nearly fulfilled (middle gray) or unfulfilled (dark gray).

| Band | RMSE | MAE | RE | $R^2$ | Band | RMSE | MAE | RE | $R^2$ |
|------|------|-----|-----|------|------|------|-----|-----|------|
| B06 | $9.28 \times 10^{-3}$ | $4.79 \times 10^{-3}$ | $2.67 \times 10^{-2}$ | $9.93 \times 10^{-1}$ | B07 | $1.07 \times 10^{-2}$ | $4.93 \times 10^{-3}$ | $2.64 \times 10^{-2}$ | $9.93 \times 10^{-1}$ |
| B07 | $1.68 \times 10^{-2}$ | $5.49 \times 10^{-3}$ | $3.47 \times 10^{-2}$ | $9.89 \times 10^{-1}$ | B8A | $1.95 \times 10^{-2}$ | $7.84 \times 10^{-3}$ | $4.34 \times 10^{-2}$ | $9.84 \times 10^{-1}$ |
| B05 | $9.31 \times 10^{-3}$ | $3.96 \times 10^{-3}$ | $5.12 \times 10^{-2}$ | $9.94 \times 10^{-1}$ | B06 | $1.01 \times 10^{-2}$ | $4.22 \times 10^{-3}$ | $4.07 \times 10^{-2}$ | $9.94 \times 10^{-1}$ |
| B05 | $6.52 \times 10^{-3}$ | $3.58 \times 10^{-3}$ | $3.88 \times 10^{-2}$ | $9.94 \times 10^{-1}$ | B11 | $1.66 \times 10^{-2}$ | $1.12 \times 10^{-2}$ | $7.57 \times 10^{-2}$ | $9.86 \times 10^{-1}$ |
| B06 | $1.52 \times 10^{-2}$ | $4.85 \times 10^{-3}$ | $3.24 \times 10^{-2}$ | $9.92 \times 10^{-1}$ | B11 | $1.75 \times 10^{-2}$ | $1.16 \times 10^{-2}$ | $9.04 \times 10^{-2}$ | $9.82 \times 10^{-1}$ |
| B06 | $4.38 \times 10^{-2}$ | $1.58 \times 10^{-2}$ | $1.00 \times 10^{-1}$ | $9.47 \times 10^{-1}$ | B8A | $1.90 \times 10^{-2}$ | $5.88 \times 10^{-3}$ | $7.06 \times 10^{-2}$ | $9.92 \times 10^{-1}$ |
| B05 | $8.20 \times 10^{-3}$ | $4.16 \times 10^{-3}$ | $3.64 \times 10^{-2}$ | $9.94 \times 10^{-1}$ | B07 | $4.34 \times 10^{-2}$ | $1.66 \times 10^{-2}$ | $1.21 \times 10^{-1}$ | $8.41 \times 10^{-1}$ |
| B05 | $3.79 \times 10^{-2}$ | $8.14 \times 10^{-3}$ | $9.38 \times 10^{-2}$ | $8.74 \times 10^{-1}$ | B12 | $1.92 \times 10^{-2}$ | $1.07 \times 10^{-2}$ | $1.27 \times 10^{-1}$ | $9.66 \times 10^{-1}$ |
| B05 | $9.64 \times 10^{-3}$ | $4.24 \times 10^{-3}$ | $3.93 \times 10^{-2}$ | $9.96 \times 10^{-1}$ | B8A | $4.63 \times 10^{-2}$ | $1.93 \times 10^{-2}$ | $1.31 \times 10^{-1}$ | $9.45 \times 10^{-1}$ |
| B07 | $1.74 \times 10^{-2}$ | $4.45 \times 10^{-3}$ | $4.05 \times 10^{-2}$ | $9.88 \times 10^{-1}$ | B11 | $5.92 \times 10^{-2}$ | $2.88 \times 10^{-2}$ | $1.97 \times 10^{-1}$ | $8.47 \times 10^{-1}$ |
| B8A | $2.66 \times 10^{-2}$ | $7.06 \times 10^{-3}$ | $5.44 \times 10^{-2}$ | $9.73 \times 10^{-1}$ | B12 | $1.64 \times 10^{-2}$ | $1.00 \times 10^{-2}$ | $2.26 \times 10^{-1}$ | $9.72 \times 10^{-1}$ |
| B07 | $1.93 \times 10^{-2}$ | $5.07 \times 10^{-3}$ | $4.74 \times 10^{-2}$ | $9.87 \times 10^{-1}$ | B12 | $5.17 \times 10^{-2}$ | $2.00 \times 10^{-2}$ | $2.37 \times 10^{-1}$ | $7.91 \times 10^{-1}$ |
| B06 | $7.30 \times 10^{-3}$ | $3.88 \times 10^{-3}$ | $3.07 \times 10^{-2}$ | $9.95 \times 10^{-1}$ | B12 | $3.63 \times 10^{-2}$ | $1.78 \times 10^{-2}$ | $2.52 \times 10^{-1}$ | $8.59 \times 10^{-1}$ |
| B8A | $5.70 \times 10^{-2}$ | $1.47 \times 10^{-2}$ | $1.25 \times 10^{-1}$ | $8.87 \times 10^{-1}$ | B11 | $6.20 \times 10^{-2}$ | $2.80 \times 10^{-2}$ | $2.79 \times 10^{-1}$ | $7.51 \times 10^{-1}$ |
| B11 | $3.73 \times 10^{-2}$ | $2.45 \times 10^{-2}$ | $2.93 \times 10^{-1}$ | $9.18 \times 10^{-1}$ | B12 | $3.26 \times 10^{-2}$ | $2.03 \times 10^{-2}$ | $2.24 \times 10^{-1}$ | $9.05 \times 10^{-1}$ |

**Table 16.** Double-band regression results for L2A. The colors in the RE (relative error) columns indicate whether the specification is fulfilled (light gray), nearly fulfilled (middle gray) or unfulfilled (dark gray).

| Band | RMSE | MAE | RE | $R^2$ | Band | RMSE | MAE | RE | $R^2$ |
|------|------|-----|-----|------|------|------|-----|-----|------|
| B5 | $7.38 \times 10^{-3}$ | $4.96 \times 10^{-3}$ | $1.77 \times 10^{-1}$ | $9.95 \times 10^{-1}$ | B8A | $9.31 \times 10^{-3}$ | $6.22 \times 10^{-3}$ | $1.38 \times 10^{-1}$ | $9.96 \times 10^{-1}$ |
| B5 | $1.11 \times 10^{-2}$ | $6.07 \times 10^{-3}$ | $1.80 \times 10^{-1}$ | $9.95 \times 10^{-1}$ | B6 | $1.33 \times 10^{-2}$ | $6.37 \times 10^{-3}$ | $1.25 \times 10^{-1}$ | $9.94 \times 10^{-1}$ |
| B6 | $8.68 \times 10^{-3}$ | $5.07 \times 10^{-3}$ | $1.91 \times 10^{-1}$ | $9.96 \times 10^{-1}$ | B12 | $1.53 \times 10^{-2}$ | $9.59 \times 10^{-3}$ | $3.52 \times 10^{-1}$ | $9.77 \times 10^{-1}$ |
| B6 | $1.24 \times 10^{-2}$ | $6.70 \times 10^{-3}$ | $5.37 \times 10^{-1}$ | $9.89 \times 10^{-1}$ | B7 | $1.54 \times 10^{-2}$ | $7.70 \times 10^{-3}$ | $1.51 \times 10^{-1}$ | $9.85 \times 10^{-1}$ |
| B5 | $1.26 \times 10^{-2}$ | $5.35 \times 10^{-3}$ | $2.03 \times 10^{-1}$ | $9.94 \times 10^{-1}$ | B11 | $1.72 \times 10^{-2}$ | $1.15 \times 10^{-2}$ | $1.94 \times 10^{-1}$ | $9.85 \times 10^{-1}$ |
| B7 | $7.30 \times 10^{-3}$ | $4.72 \times 10^{-3}$ | $1.23 \times 10^{-1}$ | $9.96 \times 10^{-1}$ | B12 | $1.74 \times 10^{-2}$ | $1.11 \times 10^{-2}$ | $1.97 \times 10^{-1}$ | $9.78 \times 10^{-1}$ |
| B7 | $1.13 \times 10^{-2}$ | $5.98 \times 10^{-3}$ | $1.10 \times 10^{-1}$ | $9.94 \times 10^{-1}$ | B8A | $1.74 \times 10^{-2}$ | $7.70 \times 10^{-3}$ | $1.65 \times 10^{-1}$ | $9.87 \times 10^{-1}$ |
| B8A | $1.21 \times 10^{-2}$ | $6.10 \times 10^{-3}$ | $1.91 \times 10^{-1}$ | $9.95 \times 10^{-1}$ | B12 | $1.79 \times 10^{-2}$ | $1.08 \times 10^{-2}$ | $3.03 \times 10^{-1}$ | $9.75 \times 10^{-1}$ |
| B8A | $1.07 \times 10^{-2}$ | $6.74 \times 10^{-3}$ | $1.09 \times 10^{-1}$ | $9.94 \times 10^{-1}$ | B11 | $1.93 \times 10^{-2}$ | $1.31 \times 10^{-2}$ | $2.22 \times 10^{-1}$ | $9.79 \times 10^{-1}$ |
| B6 | $9.79 \times 10^{-3}$ | $5.60 \times 10^{-3}$ | $1.47 \times 10^{-1}$ | $9.95 \times 10^{-1}$ | B8A | $1.99 \times 10^{-2}$ | $8.58 \times 10^{-3}$ | $2.10 \times 10^{-1}$ | $9.81 \times 10^{-1}$ |
| B11 | $4.14 \times 10^{-2}$ | $2.89 \times 10^{-2}$ | $3.29 \times 10^{-1}$ | $9.01 \times 10^{-1}$ | B12 | $3.54 \times 10^{-2}$ | $2.36 \times 10^{-2}$ | $4.86 \times 10^{-1}$ | $8.90 \times 10^{-1}$ |
| B7 | $3.74 \times 10^{-2}$ | $1.41 \times 10^{-2}$ | $5.18 \times 10^{-1}$ | $9.21 \times 10^{-1}$ | B11 | $4.98 \times 10^{-2}$ | $2.50 \times 10^{-2}$ | $2.37 \times 10^{-1}$ | $8.56 \times 10^{-1}$ |
| B5 | $5.80 \times 10^{-2}$ | $1.17 \times 10^{-2}$ | $2.11 \times 10^{-1}$ | $8.73 \times 10^{-1}$ | B12 | $1.71 \times 10^{-2}$ | $1.09 \times 10^{-2}$ | $4.36 \times 10^{-1}$ | $9.76 \times 10^{-1}$ |
| B6 | $6.55 \times 10^{-2}$ | $1.44 \times 10^{-2}$ | $7.58 \times 10^{-1}$ | $8.01 \times 10^{-1}$ | B11 | $4.06 \times 10^{-2}$ | $1.88 \times 10^{-2}$ | $3.21 \times 10^{-1}$ | $8.75 \times 10^{-1}$ |
| B5 | $7.70 \times 10^{-2}$ | $4.45 \times 10^{-2}$ | $2.87 \times 10^{-1}$ | $8.35 \times 10^{-1}$ | B7 | $9.79 \times 10^{-3}$ | $5.83 \times 10^{-3}$ | $9.24 \times 10^{-2}$ | $9.96 \times 10^{-1}$ |

For L2A data, Figure 12 presents the scatterplots for the pairs of bands in the three first rows of Table 16. As with the L1C case, the scatterplots show unbiased estimations with small dispersions, except for the B12 band in the third pair. The random sample of the test set used for generating these scatterplots does not contain pixels showing the underestimation of reflectances higher than one, but they also exist.

It is difficult to explain these results. First of all, the pairs of bands that are predicted the *best* differ between L1C and L2A. This was already the case for the regression of a single band, but in that case we could clearly define two groups: red edge–NIR and SWIR. In the case of two bands, one could have expected that for a pair of bands to be correctly reconstructed, they would have to be nonadjacent so that the missing information could be reconstructed using the neighboring bands. However, we see that the best pair in L1C is (B06, B07) and that the second best pair in L2C is (B5, B6).

With the same kind of reasoning, one could have expected that the pair (B11, B12) should be the one with the largest errors, since reconstructing the SWIR bands using only the VIS–NIR range should be nearly impossible. This is the case in terms of relative error, but not in terms of RMSE, which makes SWIR a better candidate for L2A reconstruction than more spectrally distant pairs.

Figure 13 presents the scatterplots for the prediction of the SWIR bands in L1C (top row) and L2A (bottom row). Although the dispersions are important, there is no systematic bias in the estimation, which confirms the redundancy of spectral information for most surfaces.

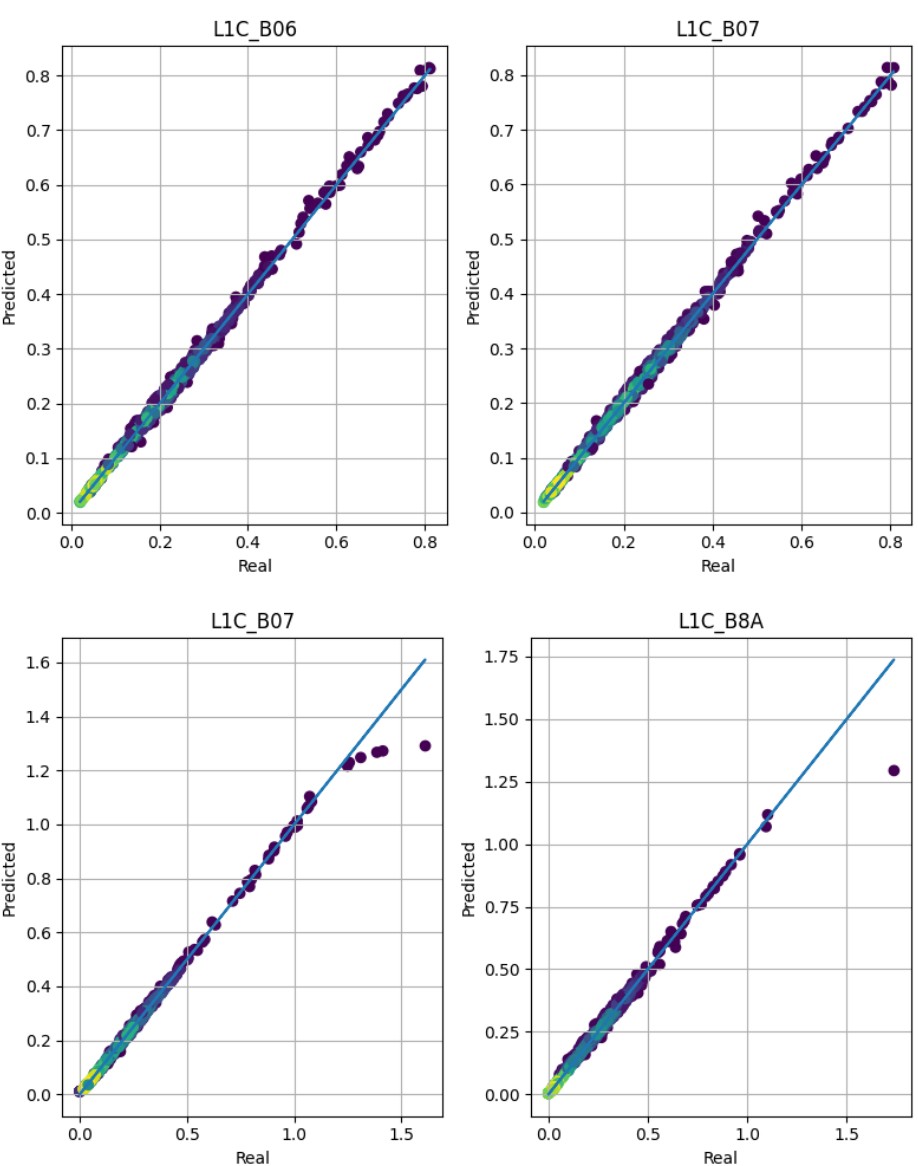

**Figure 11.** Scatterplots for the double-band regression (L1C). Each row in the figure corresponds to a row in Table 15. The colors ▬▬▬ indicate the increasing density of points.

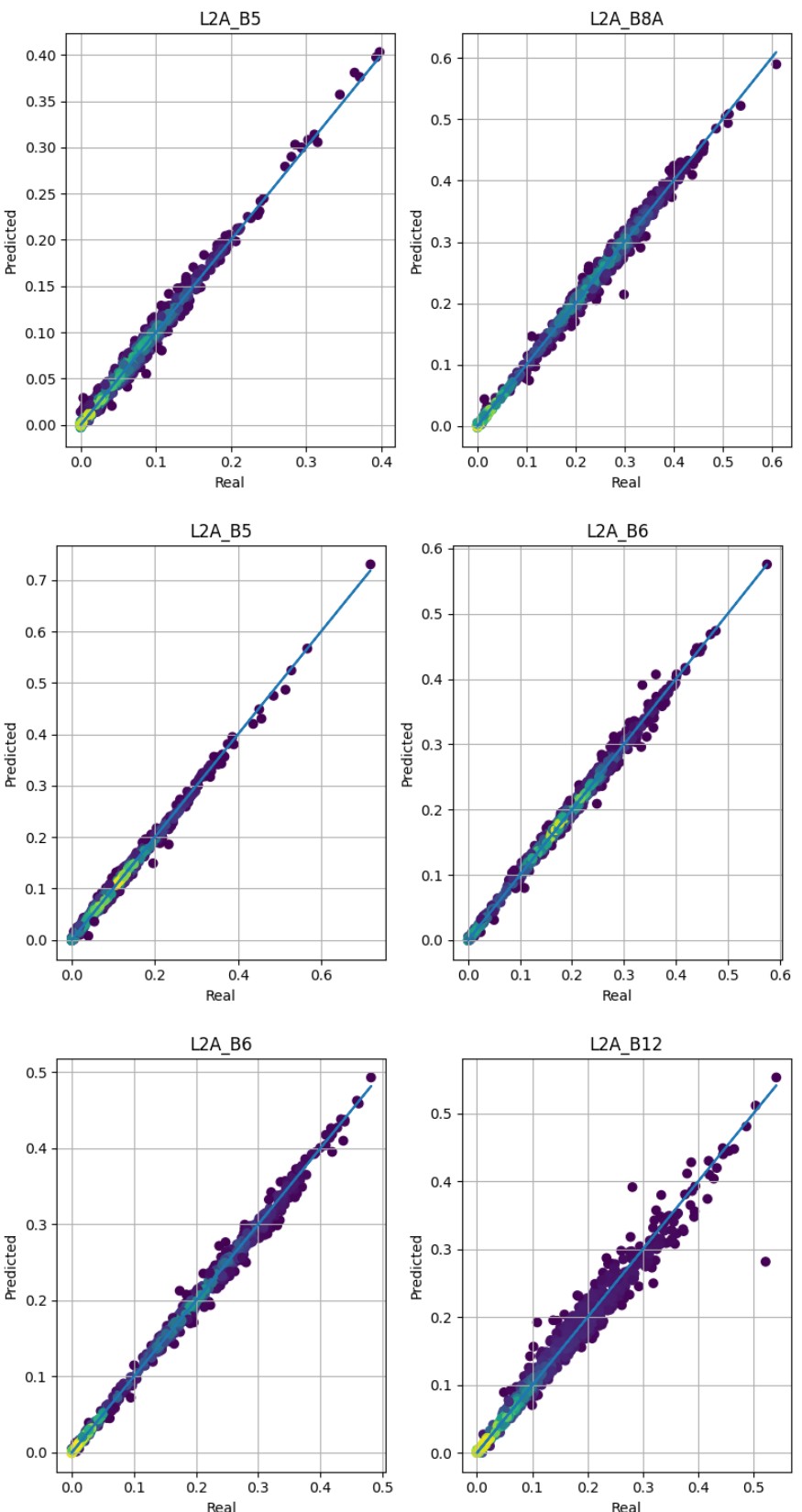

**Figure 12.** Scatterplots for the double-band regression (L2A). Each row in the figure corresponds to a row in Table 16. The colors ▬▬▬▬ indicate the increasing density of points.

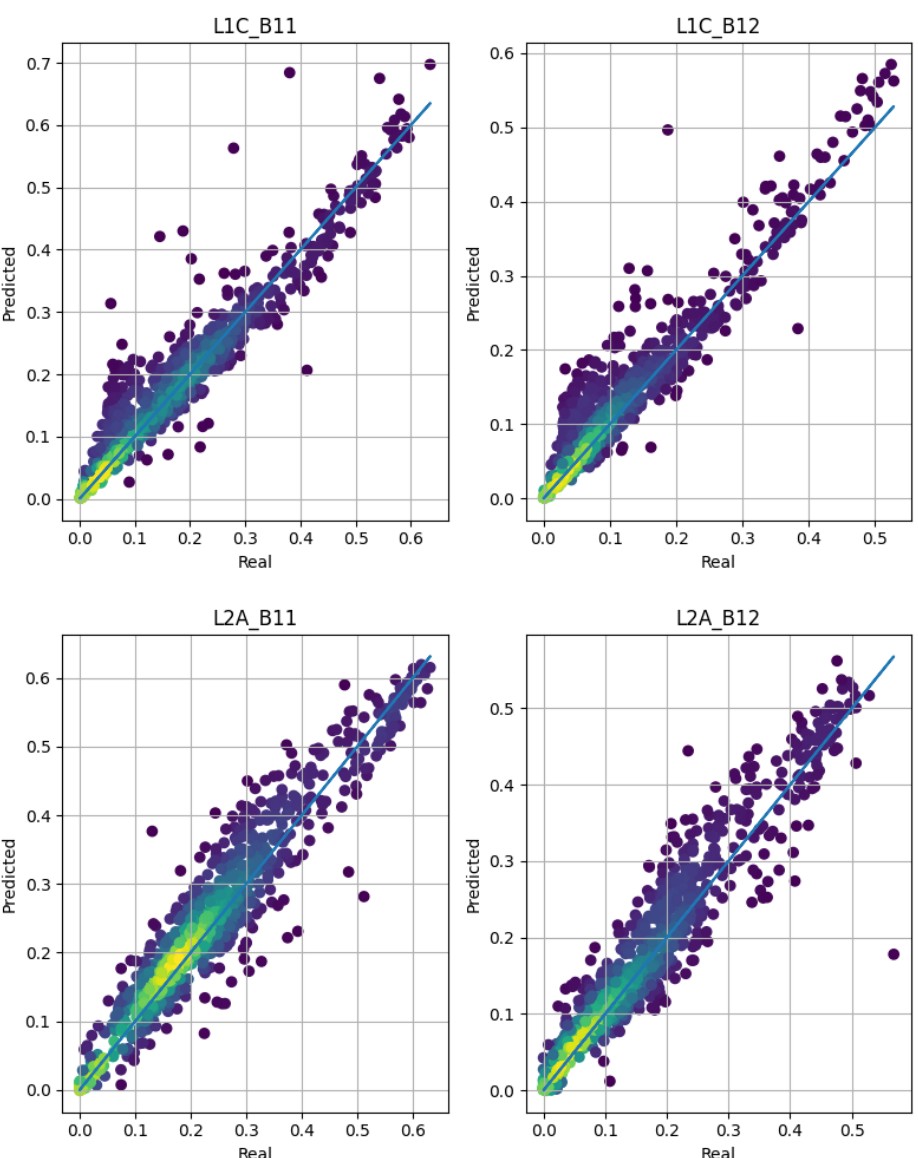

**Figure 13.** Scatterplots for the double-band regression of the SWIR bands in L1C (**top**) and L2A (**bottom**). The colors ▬▬▬▬ indicate the increasing density of points.

Actually, if the scatterplots of Figure 13 were obtained for biophysical parameter estimations, for instance LAI, chlorophyll, biomass, etc., they would be considered very good (see, for instance [30] or [18]). Of course, image quality criteria need to be more strict than those of downstream tasks, but this kind of result suggests that the impact of reflectance noise in downstream applications needs to be assessed.

## 4. Discussion

From the results presented in Section 3, we can identify several limitations of this work.

First, analysis of the errors based on type of surface (material, land cover, vegetation status, etc.) should be carried out in order to assess the impact on different applications. Although the spatial sampling of the data for this study contained enough variability for the results to be general, particular surfaces with specificities may need special attention. Furthermore, selecting the appropriate samples in the areas of most interest for particular applications can allow fine-tuning of the regression algorithm and improve the performance of the estimations.

A second limitation is related to the choice of regression algorithm for the study. The goal of the work was not to propose an optimal regression algorithm, but rather to show that band reconstruction is possible using regression. The choice of the neural network with a negative log-likelihood as a loss function was made for simplicity of implementation, the possibility of performing multi-target regression, and the generation of uncertainties associated with the estimations. Other approaches could yield better results and even produce a different set of bands candidate for removal.

All of the above suggests that replication of the study by other teams would be useful. For this purpose, the dataset has been published [25], and the source code is available for inspection and download (https://src.koda.cnrs.fr/mmdc/mmdc-legacy/-/blob/master/mmdc/spectral_regression.py, accessed on 10 May 2022).

A third limitation is the pixel-based approach taken here. Reconstructing a missing band from the reflectances of the other bands of the same pixel assumes unicity of the solution: one combination of observed bands can only correspond to one value of the missing band. Although the results of this study tend to show that this is the case, there are pixels for which the error is high. In the current setting, the regression algorithm is able to flag these pixels by reporting high uncertainty, but this is not fully satisfactory. One way of lifting the ambiguity would be to add some spatial context for the regression, so that the observations of neighboring pixels, and therefore the local texture, helps the prediction. This could be implemented with spatial convolutional layers in the regression algorithm.

In the same way, a multi-temporal extension of the algorithm could improve the estimations. However, this extension is less straightforward than adding spatial context, since clouds and cloud shadows introduce irregular temporal sampling that should be taken into account. Further, the relative geometric accuracy of multi-temporal series should be taken into account in this case.

## 5. Conclusions

In this paper, we have investigated the possibility of reconstructing one or two of the 20 m resolution bands from Sentinel-2 using the remaining bands. The goal of the study was to assess the possibility of removing some of the current Sentinel-2 spectral bands for the next generations of similar satellites.

The interest of working on band reconstruction is that the approach is independent of the application. The main rationale is, if a band can be reconstructed with errors which are within the radiometric requirements of the sensor, downstream applications can use a reconstructed band instead of a real measure.

The main findings of the study are that, at the least, one of the bands among B5, B6, B7 and B8A could be removed from next generation sensors, as all of them can be reconstructed with small errors when the others are available. Removing two bands could be possible at the cost of slightly higher reconstruction errors.

We have also shown that the estimation of a credibility interval for the predicted reflectances is possible and can therefore be used as a quality mask.

However, this study has several limitations that have been clearly identified in section 4 and that would need to be addressed in the future.

If the next generation of Sentinel-2 had one or several bands removed, one could argue that the approach presented in this work could not be applied, since the regression calibration (i.e., the neural network training) needs the target band. Several responses can be given to this argument. If the bands used as predictors remain the same in the new sensor, the regressions calibrated with the current Sentinel-2 data should be applicable.

If the bands used as predictors for this study were not available in the next generation of satellites, one could constitute enough training data by using acquisitions from a hyperspectral mission such as CHIME [31]. The appropriate spectral bands (predictors and targets) could be generated using the relative spectral responses of the next generation of Sentinel-2.

Finally, given the temporal revisit of Sentinel-2, it would be interesting to evaluate the possibility of having different bands in different satellites of the constellation, so that the band predictions could be temporally interleaved. For instance, with two satellites, one could imagine removing B6 in the A unit and removing B7 in the B unit. In this configuration, the reconstruction of B6 at a given date could use the other bands for this acquisition, as well as the most recent acquisition of the other satellite, for which B6 would have been observed. This kind of scenario would allow for interesting configurations where one satellite of the pair could have the SWIR bands absent. Indeed, the variance observed in Figure 13 could be highly reduced if the two SWIR bands for a previous date were available. Of course, for surfaces where the SWIR signature can change quickly during cloudy periods (snow falls), the impact of this kind of setting should be studied. Fortunately, available Sentinel-2 archive data allows that.

Another interesting possibility of the approaches presented in this paper is the *addition* of new bands, but only in some satellites of the constellation. On this topic, we should stress the comments on [32], as we did in Section 1.4: the fact that a particular phenomenon has a signature in a particular band does not mean that this same phenomenon cannot be detected by using a (nonlinear) combination of other bands. The results presented in this paper indicate that the question can be reversed.

The attentive reader will have understood that many options are open to reduce costs and hardware complexity for the successors of the current Sentinel-2 system by leveraging spectral, spatial and temporal correlations of the observed surfaces through ground data processing.

This work is just an example of what could be done by using the richness of the Sentinel-2 archives. We think that with the help of other scientists, further studies could be defined. For instance, a subset of geographic areas and dates for each target application together with ground measures could be made available. This would allow the objective assessment of errors due to the lack of particular bands.

**Author Contributions:** Conceptualization, J.I. and O.H.; methodology, J.I.; software, J.I. and J.M.; validation, J.I.; formal analysis, J.I. and O.H.; investigation, J.I.; resources, J.I., J.M. and O.H.; data curation, J.I.; writing—original draft preparation, J.I.; writing—review and editing, J.I., J.M. and O.H.; visualization, J.I.; supervision, J.I.; project administration, J.I.; funding acquisition, J.I. and O.H. All authors have read and agreed to the published version of the manuscript.

**Funding:** This research received no external funding.

**Data Availability Statement:** The dataset has been published [25], and the source code is available for inspection and download at https://src.koda.cnrs.fr/mmdc/mmdc-legacy/-/blob/master/mmdc/spectral_regression.py, accessed on 10 May 2022.

**Conflicts of Interest:** The authors declare no conflict of interest.

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
