# Peer review of "Assessment of the Usefulness of Spectral Bands for the Next Generation of Sentinel-2 Satellites by Reconstruction of Missing Bands"

_remotesensing, doi:10.3390/rs14102503_

Round 1

Reviewer 1 Report

Excellent work!

My only comment is related to the not so recent references used in the manuscript.

Reviewer 2 Report

A useful paper on the relevance of Sentinel2 bands for compositional mapping. A significant contribution for discussions of future Sentinel mission specifications. However the manuscript lacks comments and references on the geoscience applications (eg Table 2's limitation to vegetation indices) and the implications for S2 band redundancy (see pdf comments). Some acronyms are also lacking definition.

Reviewer 3 Report

This manuscrip is well written, has a really deep introduction presenting the interest of the study.

The material and methods and results are clearly described.

THe conclusion is clear and presents interesting options to new studies in the future.

My only negative points concerns the form of the manuscript.

I appreciate the novelty of the way it's written, but it will be better if the section Material and methods is clearly identified.

There are some methods which are described in the results section and figure 5 should be moved to the results section.

Tables 5 to 14 could be grouped (not all together but at least reduce the number of tables to facilitate the lecture and maybe find a color legend to identify quicker when the prediction is very good, good, pass and bad for each band.

Tables 15 and 16 good number

Some figure legends could be enriched (specially explaining the colors) with more data to make them independent from the text.

Please find the in attached document the identified parts which should be changed
